# Blockade of the SRC/STAT3/BCL-2 Signaling Axis Sustains the Cytotoxicity in Human Colorectal Cancer Cell Lines Induced by Dehydroxyhispolon Methyl Ether

**DOI:** 10.3390/biomedicines11092530

**Published:** 2023-09-13

**Authors:** Ya-Chu Hsieh, Yuan-Chang Dai, Kur-Ta Cheng, Wei-Ting Yang, Modukuri V. Ramani, Gottumukkala V. Subbaraju, Yi-Ju Chen, Chia-Che Chang

**Affiliations:** 1Doctoral Program in Tissue Engineering and Regenerative Medicine, National Chung Hsing University, Taichung 402202, Taiwan; asdf789515@gmail.com; 2Department of Pathology, Ditmanson Medical Foundation Chia-Yi Christian Hospital, Chiayi 600566, Taiwan; cych03884@gmail.com; 3Department of Laboratory Medicine, Ditmanson Medical Foundation Chia-Yi Christian Hospital, Chiayi 600566, Taiwan; 4Doctoral Program in Translational Medicine, National Chung Hsing University, Taichung 402202, Taiwan; 5Department of Biochemistry and Molecular Cell Biology, Taipei Medical University, Taipei 110301, Taiwan; ktbot@tmu.edu.tw; 6Department of Life Sciences, National Chung Hsing University, Taichung 402202, Taiwan; bunnytw6@gmail.com; 7Department of Organic Chemistry, Andhra University, Visakhapatnam 530003, India; ramani_v@yahoo.com (M.V.R.); subbarajugv@gmail.com (G.V.S.); 8Department of Dermatology, Taichung Veterans General Hospital, Taichung 407219, Taiwan; 9Department of Post-Baccalaureate Medicine, National Chung Hsing University, Taichung 402202, Taiwan; 10Graduate Institute of Biomedical Sciences, Doctoral Program in Translational Medicine, Rong Hsing Research Center for Translational Medicine, The iEGG and Animal Biotechnology Research Center, National Chung Hsing University, Taichung 402202, Taiwan; 11Department of Medical Laboratory Science and Biotechnology, Asia University, Taichung 413305, Taiwan; 12Department of Medical Research, China Medical University Hospital, Taichung 404333, Taiwan; 13Traditional Herbal Medicine Research Center, Taipei Medical University Hospital, Taipei 110301, Taiwan

**Keywords:** dehydroxyhispolon methyl ether, hispolon, STAT3, SRC, BCL-2, apoptosis, cytotoxicity, colorectal cancer

## Abstract

Colorectal cancer (CRC) is the third most prevalent human cancer globally. 5-Fluorouracil (5-FU)-based systemic chemotherapy is the primary strategy for advanced CRC treatment, yet is limited by poor response rate. Deregulated activation of signal transducer and activator of transcription 3 (STAT3) is fundamental to driving CRC malignant transformation and a poor prognostic marker for CRC, underscoring STAT3 as a promising CRC drug target. Dehydroxyhispolon methyl ether (DHME) is an analog of Hispolon, an anticancer polyphenol abundant in the medicinal mushroom *Phellinus linteus*. Previously, we have established DHME’s cytotoxic effect on human CRC cell lines by eliciting apoptosis through the blockade of WNT/β-catenin signaling, a preeminent CRC oncogenic pathway. Herein, we unraveled that compared with 5-FU, DHME is a more potent killer of CRC cells while being much less toxic to normal colon epithelial cells. DHME suppressed both constitutive and interleukin 6 (IL-6)-induced STAT3 activation represented by tyrosine 705 phosphorylation of STAT3 (p-STAT3 (Y705)); notably, DHME-induced CRC apoptosis and clonogenicity limitation were abrogated by ectopic expression of STAT3-C, a dominant-active *STAT3* mutant. Additionally, we proved that BCL-2 downregulation caused by DHME-mediated STAT3 blockage is responsible for DHME-induced CRC cell apoptosis. Lastly, DHME inhibited SRC activation, and v-src overexpression restored p-STAT3 (Y705) levels along with lowering the levels of apoptosis in DHME-treated CRC cells. We conclude DHME provokes CRC cell apoptosis by blocking the SRC/STAT3/BCL-2 axis besides thwarting WNT/β-catenin signaling. The notion that DHME targets two fundamental CRC signaling pathways underpins the potential of DHME as a CRC chemotherapy agent.

## 1. Introduction

Colorectal cancer (CRC) remains the third most prevalent human malignancy and represents a leading cause of cancer-related lethality worldwide in the latest GLOBOCAN estimates [1]. Surgical resection is adequate for early-stage CRC, while systemic chemotherapy remains the primary strategy for metastatic CRC treatment [2]. 5-Fluorouracil (5-FU) is the most effective and commonly used drug for CRC chemotherapy, but the overall response rate of 5-FU-based regimens in advanced CRC is lower than 20% due to acquired chemoresistance [3,4,5]. Alcohol consumption, low physical activity, obesity, and smoking are recognized as lifestyle-related risk factors for CRC development [6]. In addition, the dysregulation of signaling pathways such as EGFR/MAPK, Hedgehog, Notch, PI3K/AKT, TGF-β, or WNT/β-catenin is fundamental to driving CRC carcinogenesis and malignant progression [7,8]. It appears that the WNT/β-catenin signaling is preeminent among these CRC-driving pathways, as evidenced by the oncogenic WNT/β-catenin activation in almost all CRC cases [9,10,11]. Recently, aberrant activation of signal transducer and activator of transcription 3 (STAT3) has emerged as a vital driver for CRC development [12].

STAT3 is a transcription factor integral to regulating diverse physiological processes such as cell survival, proliferation, differentiation, and immune responses [13]. Activation of STAT3 is primarily achieved via the phosphorylation of STAT3 at tyrosine 705 residue by upstream tyrosine kinases like Janus kinases (JAK) and SRC to mediate the signals incited by cytokines (e.g., interleukin 6 (IL-6)) and growth factors, respectively. Tyrosine 705-phosphorylation of STAT3 allows the homodimerization and subsequent nuclear translocation of STAT3 to activate the transcription of target genes such as *BCL-2*, *BCL-xL*, and *MCL-1* for modulating biological outcomes [14,15]. Although the physiological activation of STAT3 is rapid and transient, dysregulated STAT3 activation is generally found in a broad range of human cancers and is essential to driving malignant transformation and progression [16,17]. Concerning CRC, persistent STAT3 activation has been linked to enhanced CRC cell proliferation and tumor growth [18] and is correlated with poor prognosis of clinical outcomes for CRC patients [19,20]. Notably, numerous preclinical studies have confirmed the pharmacological blockade of STAT3 as a promising strategy for CRC therapy, supporting the role of STAT3 as a potential target for developing novel CRC therapeutics [21,22,23,24].

Natural compounds are invaluable resources for discovering novel therapeutics, including anticancer agents [25,26]. Hispolon is a bioactive polyphenolic compound present in the medicinal mushroom *Phellinus linteus*. Preclinical studies have established Hispolon’s antiproliferative effects on diverse human cancer cell lines in vitro and in vivo [27,28], supporting Hispolon as a potential lead compound for designing novel anticancer drugs [29,30,31]. Dehydroxyhispolon methyl ether (DHME) is one of the Hispolon analogs derived from chemical modifications of Hispolon’s structure [30]. Our previous report initially unraveled DHME’s cytotoxic effect on human CRC cell lines while sparing normal human colon epithelial cells; it clarified apoptosis as a primary cause of DHME’s cytotoxic action through the blockade of the WNT/β-catenin signaling [32]. In the current study, we compared the cytotoxic potency of DHME with that of 5-FU and further excavated additional mechanisms of DHEM’s cytotoxic action. Herein, for the first time, we revealed that DHME-induced cytotoxicity is more potent and selective than 5-FU against CRC cells and is exerted by blocking the STAT3-mediated signaling. Our findings of DHME’s inhibitory action on both STAT3 and WNT/β-catenin signaling axes, two fundamental signaling axes involved in CRC genesis and progression, underpin DHME as a potential agent for CRC chemotherapy.

## 2. Materials and Methods

### 2.1. Chemicals

DHME was chemically synthesized and prepared as documented by Fan et al. [32]. 5-FU was obtained from Cayman Chemical (Ann Arbor, MI, USA), prepared as a 100 mM stock solution in dimethyl sulphoxide (DMSO), and then stored at −20 °C until use. Recombinant human IL-6 was bought from PeproTech (Rehovot, ISR) and prepared as a 10 mg/mL stock solution in 1× phosphate buffered saline (PBS) (VWR International; Radnor, PA, USA). Both DMSO and polybrene were acquired from Sigma-Aldrich (St. Louis, MO, USA).

### 2.2. Cell Culture

Normal human colon epithelial cell line CCD 841 CoN (ATCC CCL-1790) was procured from the American Type Culture Collection (ATCC) (Manassas, VA, USA) and grown in Eagle’s Minimum Essential Medium (MEM) medium. Human colorectal adenocarcinoma (CRC) cell lines HCT-15 (ATCC CCL-225) and LoVo (ATCC CCL-229) were purchased from the Bioresource Collection and Research Center (Hsinchu, TWN) and cultured in RPMI-1640 and F-12K media, respectively. All culture media were replenished with 10% fetal bovine serum (FBS) and 1% penicillin–streptomycin (P/S). Besides FBS and P/S, the HCT-15 culture media were further supplemented with 1% sodium pyruvate, 1% glucose, and HEPES. All cell lines were grown at 37 °C in a humidified environment with 5% CO_2_. All chemicals applied to cell culture were acquired from Gibco Life Technologies (Carlsbad, CA, USA).

### 2.3. In Vitro Cytotoxicity Assay

CellTiter 96^®^ AQueous One Solution Cell Proliferation Assay (MTS) assay (Promega) was employed to evaluate the short-term in vitro cytotoxicity elicited by DHME or 5-FU as previously reported [33]. Briefly, CCD 841 CoN and CRC cell lines (7 × 10^3^ cells/well) were grown in 96-well culture plates for 24 h prior to drug treatment and then subjected to 24 h and 48 h drug treatments, followed by cell-viability determination in accordance to manufacturer’s protocol (Madison, WI, USA). Clonogenicity assay was utilized to examine the long-term in vitro cytotoxicity, which assesses the effect of drugs on cells’ ability to form colonies. In short, cells (4 × 10^5^) were treated with DHME (0, 20, 40 μM) for 24 h, and then 2 × 10^2^ of DHME-treated cells were seeded onto 6-well plates to grow for 10~14 days to form colonies in drug-free culture media. Colonies were exposed by staining with 1% crystal violet solution for subsequent number scoring as described previously [33].

### 2.4. Annexin V/Propidium Iodide Dual Staining Assay

The extent of DHME-induced apoptosis in human CRC cells was determined using Muse^®^ Annexin V & Dead Cell Assay Kit (Millipore; Burlington, MA, USA) for the levels of cell surface-exposed Annexin V according to our reported protocol [33]. In brief, CRC cells (3 × 10^5^ cells/well) grown on 6-well plates were treated for 24 h with DHME (0, 20, 40 μM), followed by resuspension by trypsinization and wash twice with 1× PBS. Then, cells were exposed to 100 μL of Annexin V & Dead Cell reagent for 20 min at room temperature in the dark, followed by flow cytometry analysis to score the levels of Annexin V-positive (apoptotic) cell population on the Muse^®^ Cell Analyzer (Millipore; Burlington, MA, USA).

### 2.5. Stable Clone Establishment

The pBabe-based vectors for ectopic expression of N-terminal hemagglutinin epitope (HA)-tagged dominant-active *STAT3* mutant (STAT3-C), *BCL-2*, and N-terminal HA-tagged *v-src* were respectively coined as pBabe-HA-STAT3-C, pBabe-BCL-2, and pBabe-HA-v-src and have been described previously [32,33,34]. To generate pBabe-derived retroviral particles, 2.5 μg of the pBabe-based plasmids were transfected into 293-T cells (70–80% confluency) along with the plasmids expressing gag-pol (2.5 μg) and VSVG proteins (0.5 μg) to assemble retroviral particles. The retroviral particles released into the culture media 24 h and 48 h after transfection were harvested by centrifugation (11,000× *g*) at 4 °C for 3 min to collect the supernatants, which were then used to infect HCT-15 and LoVo cells in the presence of polybrene (8 μg/mL) for infection efficiency improvement. Two days after infection, cells were subjected to positive selection by puromycin (2 μg/mL) for 48 h, followed by immunoblotting to confirm the ectopic expression of HA-STAT3-C, BCL-2, or HA-v-src.

### 2.6. Immunoblottinghme

Immunoblotting was executed based on our reported protocol [33,34]. Primary antibodies against cleaved PARP (#9541), HA-tag (#3724), phospho-STAT3 (Y705) (#9145), phospho-JAK2 (Y1007/1008) (#3776), JAK2 (#3230), phospho-Src (Y416) (#6743), and Src (#2108) were purchased from Cell Signaling Technology (Boston, MA, USA). Anti-Bcl-2 (GTX100064), anti-GAPDH (GTX110118), and anti-STAT3 (GTX104616) antibodies were obtained from GeneTex (Irvine, CA, USA). All secondary antibodies were obtained from Jackson ImmunoResearch Laboratories (West Grove, PA, USA).

### 2.7. Statistical Analysis

All data from six to nine independent experiments were presented as the mean ± standard deviation. Statistical evaluation was performed by the Kruskal–Wallis test for multiple comparisons. All calculations were conducted using GraphPad Prism 9 software (GraphPad, San Diego, CA, USA). A probability value (*p*) lower than 0.05 was considered statistically significant.

## 3. Results

### 3.1. DHME Is More Potent and Selective Than 5-FU to Exert CRC Cytotoxicity

Considering that 5-FU is an indispensable CRC chemotherapy drug, we herein compared the potency of CRC cytotoxicity between DHME and 5-FU. It is noted that human CRC cell lines HCT-15 and LoVo cells were susceptible to the cytotoxicity provoked by both drugs after 48 h treatment. The IC_50_ of DHME for HCT-15 and LoVo cells was 11.79 ± 1.05 μM and 10.42 ± 1.14 μM, respectively (Figure 1, left panel), whereas it took 87.59 ± 1.83 μM and 196.56 ± 1.25 μM of 5-FU to reach the same levels of cytotoxicity against each corresponding cell line (Figure 1, right panel). It is also noteworthy that DHME was much less toxic to normal human colon epithelial cell line CCD 841 CoN compared with its CRC cytotoxicity (Figure 1, left panel); however, 5-FU was cytotoxic to CCD 841 CoN cells at its effective dosage against CRC cells (Figure 1, right panel). These results together indicated that, in our experimental settings, DHME appeared to be more potent and selective than 5-FU regarding the induction of CRC cytotoxicity.

### 3.2. DHME Suppresses Both Constitutive and Inducible STAT3 Activation in CRC Cells

Persistent activation of STAT3 is another critical oncogenic driver for CRC genesis besides aberrant WNT/β-catenin signaling. In view of that, we addressed the effect of DHME on the status of STAT3 activation, which can be revealed by immunoblotting for the level of phosphorylation at the tyrosine 705 residue of STAT3 (p-STAT3 (Y705)) [12,13]. It is noticed that DHME reduced p-STAT3 (Y705) levels in HCT-15 and LoVo cells in a dose- and time-dependent manner, illustrating that DHME suppressed the constitutive STAT3 activation inherently present in CRC cells (Figure 2A,B and Appendix A). We then asked whether DHME can suppress STAT3 activation induced by external stimuli such as interleukin-6 (IL-6). We observed that IL-6 treatment led to a marked elevation of p-STAT3 (Y705) levels, as expected, whereas IL-6-induced p-STAT3 (Y705) upregulation was abrogated when co-treated with DHME (Figure 2C and Appendix A). These lines of evidence confirm the inhibitory action of DHME on STAT3 activation in CRC cells.

### 3.3. Blockade of STAT3 Activation Is Required for DHME-Induced CRC Cytotoxicity

We next inquired the functional significance of STAT3 blockage in DHME-elicited cytotoxicity. To address this, we established CRC cell clones with stable ectopic expression of a dominant-active mutant of *STAT3* (STAT3-C) [35] to counteract DHME’s inhibitory action on STAT3. Immunoblotting revealed that in HCT-15 and LoVo cells, DHME markedly induced apoptosis in vector control clones as evidenced by the increase in the levels of PARP cleavage, an apoptotic marker [36]; in contrast, DHME-induced PARP cleavage was dramatically abolished in STAT3-C stable clones (Figure 3A and Appendix A). The resistance of STAT3-C stable clones to DHME-induced apoptosis was further validated by a marked reduction in the levels of Annexin V-positive (hence apoptotic) cell population in DHME-treated STAT3-C stable clones when compared with those of vector control clones (Figure 3B). Specifically, the apoptotic population elicited by 20 μM of DHME was reduced from 64.55 ± 3.27% in vector control clones to 37.35 ± 6.97% in STAT3-C clones (*p* < 0.001). Likewise, when treated with 20 μM of DHME, 70.23 ± 3.14% of LoVo vector control clone underwent apoptosis, which was lowered to 43.18 ± 6.01% in its corresponding STAT3-C clones (*p* < 0.001). Moreover, the decrease in apoptosis corresponds to the increase in the clonogenicity of STAT3-C stable clones after DHME treatment (Figure 3C). Thus, our finding that ectopically sustained STAT3 activation curtailed DHME’s cytotoxic action supports that inhibition of STAT3-mediated signaling is an integral mechanism of action whereby DHME exerts its CRC cytotoxicity.

### 3.4. DHME Downregulates BCL-2 by Curbing STAT3 Activation to Induce CRC Cytotoxicity

To further substantiate the role of STAT3-mediated signaling in the cytotoxic action of DHME, we examined the effect of DHME on the downstream effectors of the STAT3 signaling pathway such as the antiapoptotic BCL-2 family members BCL-2, BCL-xL, and MCL-1. Immunoblotting unraveled that DHME showed limited effect on BCL-xL; nevertheless, it obviously downregulated BCL-2 while upregulating MCL-1 (Figure 4A and Appendix A). Notably, DHME-induced BCL-2 downregulation was sabotaged when STAT3 remained active, as evidenced by the restoration of BCL-2 levels in DHME-treated STAT3-C stable clones (Figure 4B and Appendix A). This finding strongly argues that DHME inhibits STAT3 activation to lower the expression levels of BCL-2. We next addressed whether BCL-2 downregulation mediates the cytotoxic effect of DHME on CRC cells. To this end, we generated CRC cell clones with stable BCL-2 overexpression to evaluate DHME-induced cytotoxicity. Immunoblotting revealed that DHME evidently increased the levels of cleaved PARP (c-PARP) in vector control clones while failing to promote PARP cleavage when BCL-2 was not downregulated (Figure 4C and Appendix A), proving that DHME lowers BCL-2 levels to induce CRC cell apoptosis. In consistence, we also observed that CRC cells was more refractory to DHME-evoked increase in Annexin V-positive (i.e., apoptotic) population when BCL-2 was overexpressed (Figure 4D). Overall, these results indicated that DHME inhibits STAT3 activation to downregulate BCL-2, leading to the induction of apoptosis to kill CRC cells.

### 3.5. DHME Inhibits SRC to Suppress the Activation of STAT3 in CRC Cells

Lastly, we aimed to elucidate how DHME inhibits STAT3 activation. We began by examining DHME’s effect on Janus kinase 2 (JAK2), a kinase well known for direct phosphorylation of STAT3 to activate STAT3. We found that DHME at 20 μM lowered the levels of activated JAK2 (i.e., JAK2 with dual phosphorylation at tyrosine residues 1007 and 1008 (p-JAK2)) in LoVo cells while mildly altering the p-JAK2 levels in HCT-15 cells (Figure 5A and Appendix A). We next explored DHME’s effect on SRC, another upstream kinase responsible for STAT3 activation. It is noteworthy that DHME reduced the levels of activated SRC (i.e., tyrosine 406-phosphorylated SRC (p-SRC)), proving DHME’s inhibitory effect on SRC activation (Figure 5B and Appendix A). To further verify that DHME blocks STAT3 activation through the inhibition of SRC, we generated stable CRC cell clones with ectopic expression of chicken v-src, a persistently active SRC mutant [37]. Our results indicated that although the levels of p-STAT3 (Y705) in DHME-treated vector control clones were reduced, DHME barely lowered p-STAT3 (Y705) levels in the v-src stable clones (Figure 5C and Appendix A). Hence, this evidence supports that inhibition of SRC represents a mechanism whereby DHME represses STAT3 activation. Moreover, we noticed that aside from restoring p-STAT3 (Y705) levels, ectopic v-src expression prevented CRC cells from DHME-elicited PARP cleavage (Figure 5C). Collectively, we conclude that DHME thwarts the activation of SRC to inhibit STAT3 activation, leading to BCL-2 downregulation for inducing apoptosis to suppressing CRC cell viability.

## 4. Discussion

Using HCT-15 and LoVo cell lines as the in vitro CRC model in the present study, we revealed that DHME is more potent and selective than 5-FU to provoke CRC cell death. We further identified that targeting the SRC/STAT3/BCL-2 signaling axis is an integral mechanism of action underlying the cytotoxic effect of DHME on CRC cells. These notions are supported by the following lines of evidence. First, the IC_50_ of DHME for CRC cells is generally below 20 μM, nearly 40- to 100-fold less than the IC_50_ of 5-FU, and also relatively non-toxic to normal colon epithelial cells, whereas 5-FU already kills more than 50% of normal colon cells at its IC_50_ dosage for CRC cells (Figure 1). Furthermore, we confirmed the inhibitory effect of DHME on both constitutive and IL6-induced activation of STAT3 (Figure 2), which is required for DHME to exert CRC cytotoxicity (Figure 3). Moreover, we proved that DHME inhibits STAT3 activation to downregulate BCL-2, leading to the induction of CRC cell apoptosis (Figure 4). Finally, we validated that DHME suppresses STAT3 activation by targeting SRC-mediated activating phosphorylation of STAT3 (Figure 5). To the best of our knowledge, the findings about the functional linkage between DHME-induced cytotoxicity and the SRC/STAT3/BCL-2 signaling axis, in addition to the distinct CRC cytotoxicity between DHME and 5-FU, are reported for the first time.

5-FU remains the central constituent of chemotherapy regimens for current CRC treatment [38]. Still, the application of 5-FU to CRC therapy is limited due to the acquired resistance frequently developed in tumor tissues after long-term treatment, besides toxic side effects to normal tissues such as the heart and liver [39,40]. Hence, novel anticancer agents are in demand to complement 5-FU-based CRC chemotherapy regimens for better therapeutic efficacy and less toxic side effects. Notably, the STAT3 signaling axis has been highlighted as a fundamental driver of acquired 5-FU resistance in CRC cells. For instance, Zhang et al. reported that the p-STAT3 (Y705)-containing exosomes derived from 5-FU-resistant RKO cells endow parental RKO cells’ resistance to 5-FU-induced cytotoxicity [41]. In contrast, Yang et al. revealed that Enalapril, an antihypertensive drug, downregulates NF-κB/STAT3-regulated proteins to overcome 5-FU resistance in CRC cells, leading to an enhanced therapeutic efficacy of 5-FU against CRC in vivo [42]. Additionally, Yue et al. elucidated that STAT3 drives 5-FU resistance in CRC cells by upregulating MCL-1 to trigger cytoprotective autophagy [43]. Herein, we revealed that DHME, compared with 5-FU, slays CRC cells at a much lower dosage while being less toxic to normal colon epithelial cells (Figure 1) and functions as a potent repressor of the STAT3 signaling axis (Figure 2, Figure 3 and Figure 4). Along this line, DHME appears as a potential candidate to be included in 5-FU-based CRC chemotherapeutic regimens to potentiate 5-FU’s therapeutic efficacy by impairing STAT3-elicited 5-FU resistance.

Intriguingly, we observed that DHME markedly upregulates MCL-1 in CRC cells (Figure 4). MCL-1 is a potent antiapoptotic BCL-2 family protein whose upregulation is frequently observed in multiple human malignancies to promote tumorigenesis and drug resistance [44]. In CRC, accumulating evidence has validated that the blockade of MCL-1 activity through pharmacological inhibition or degradation overcomes drug resistance and facilitates sensitization to targeted therapeutics [43,45,46,47,48,49]. Thus, it is plausible to speculate that DHME-induced MCL-1 upregulation would likely confer DHME resistance on CRC cells. Along this line, DHME’s efficacy of CRC cytotoxicity and 5-FU sensitization should be potentially reinforced when MCL-1 is sabotaged. In our follow-up studies, we are addressing these questions and elucidating how DHME upregulates MCL-1 in CRC cells.

Data presented here argue that the blockade of SRC is the underlying mechanism by which DHME represses STAT3 activation (Figure 5). Notably, accumulating evidence has linked deregulated SRC activity or expression levels to malignant progression, cancer stemness, and chemoresistance in CRC [50,51,52,53]. A recent report by Ruiz-Saenz et al. demonstrated that in BRAF^V600E^ colorectal tumors, the compensatory SRC activation is the driving force behind the development of resistance to the inhibitors against BRAF and epidermal growth factor receptor (EGFR) after long-term management [54]. Considering the importance of SRC in promoting CRC progression and drug resistance, our discovery of DHME acting as an inhibitor of SRC activation further implicates the potential to translate DHME into a therapeutic strategy for CRC treatment.

The current study revealed that DHME exerts in vitro CRC cytotoxicity by repressing the SRC/STAT3/BCL-2 axis. Notably, our previous report unraveled the WNT/β-catenin/BCL-2 signaling blockade as DHME’s other mechanism of cytotoxic action against CRC cells [32]. Hence, it appears that DHME provokes CRC cell death by targeting multiple antiapoptotic signaling pathways. Intriguingly, many lines of evidence have demonstrated the functional crosstalk between the STAT3 and WNT/β-catenin signaling pathways. For instance, Chen et al. uncovered that STAT3 sustains WNT/β-catenin signaling to maintain ovarian cancer stemness [55]. Furthermore, Lee et al. identified that STAT3 activation-mediated activation of WNT/β-catenin signaling accounts for the acquired resistance of non-small cell lung cancer cells to the anticancer agent 17-AAG, an Hsp90 inhibitor [56]. On the other hand, Fragoso et al. showed that STAT3 activation elicited by WNT/β-catenin signaling is required for WNT/β-catenin-mediated survival of human retinal pigment epithelial cells under oxidative stress [57]. Likewise, Kim et al. reported that WNT1 overexpression in gastric cancer cells upregulates p-STAT3 (Y705) and p-STAT3 (Y705) nuclear accumulation in a β-catenin-dependent manner and further proved that galectin-3 is responsible for bridging the WNT/β-catenin and STAT3 signaling pathways, culminating in gastric tumor progression [58]. Along this line, it would be interesting to clarify whether DHME suppresses the STAT3 and WNT/β-catenin axis by targeting the individual pathway per se or by disrupting the crosstalk between these two signaling axes.

Our current study verified DHME’s selective cytotoxicity against CRC cell lines and the SRC/STAT3/BLC-2 axis blockade as the mechanism of DHME’s cytotoxic action. Yet, it should be noted that the present lines of evidence were derived from an in vitro assessment of DHME’s cytotoxicity on CRC cells. An in vivo demonstration of DHME-induced retardation of CRC tumor growth and reduction in tyrosine 705 phosphorylation of STAT3 within CRC tumors is fundamental to underpinning DHME’s potential as a therapeutic option for CRC treatment. Accordingly, the in vivo validation of DHME’s anti-CRC action, along with the toxicity and pharmacokinetics profile, will be our primary goals in future studies.

In conclusion, the present study unraveled that DHME is more cytotoxic than 5-FU to CRC cells while sparing normal colon epithelial cells. Mechanistically, DHME targets the SRC/STAT3/BCL-2 axis for inhibition to induce CRC cell apoptosis, in addition to blocking the WNT//β-catenin signaling reported previously. Our findings implicate the potential application of DHME as a novel constituent of chemotherapy regimens for CRC treatment.

## Figures and Tables

**Figure 1 biomedicines-11-02530-f001:**
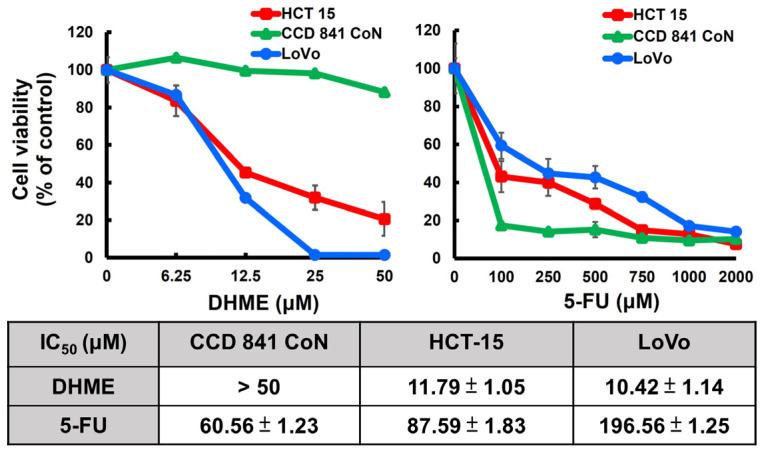
DHME was more potent and selective than 5-FU to elicit CRC cytotoxicity. Human CRC cell lines HCT-15 and LoVo and normal human colorectal epithelial cell line CCD 841 CoN were subject to 48 h treatment with graded dosage of DHME (**Left**) or 5-FU (**Right**), followed by assessment of cell viability using MTS assay. It appears that 5-FU is cytotoxic to both CRC cell lines and normal colon epithelial cells, whereas DHME kills CRC cells at much lower dosage than 5-FU while also sparing normal colon epithelial cells.

**Figure 2 biomedicines-11-02530-f002:**
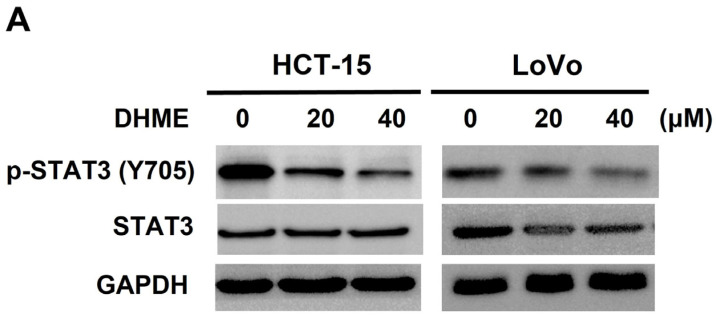
Inhibitory effect of DHME on STAT3 activation in human CRC cells. (**A**) Dose-dependent suppression of constitutive STAT3 activation by DHME. Human CRC cell lines were treated with increasing dosage of DHME (0, 20, 40 μM) for 24 h, followed by immunoblotting for the levels of tyrosine 705-phosphorylated STAT3 (p-STAT3 (Y705)), a surrogate marker of STAT3 activation. (**B**) Time-dependent suppression of constitutive STAT3 activation by DHME. Human CRC cell lines were subject to 24 h treatment with 40 μM of DHME, followed by immunoblotting for the levels of p-STAT3 (Y705) at indicated time points. (**C**) Suppression of IL-6-induced STAT3 activation by DHME. Human CRC cell lines were pre-stimulated without or with IL-6 (100 ng/mL) for 30 min, followed by treatment with DMSO or DHME (0, 20, 40 μM) for 24 h. Immunoblotting was conducted thereafter to determine the levels of p-STAT3 (Y705) in each experimental setting. In all immunoblot analyses, the amount of GAPDH was used as the control for equal loading.

**Figure 3 biomedicines-11-02530-f003:**
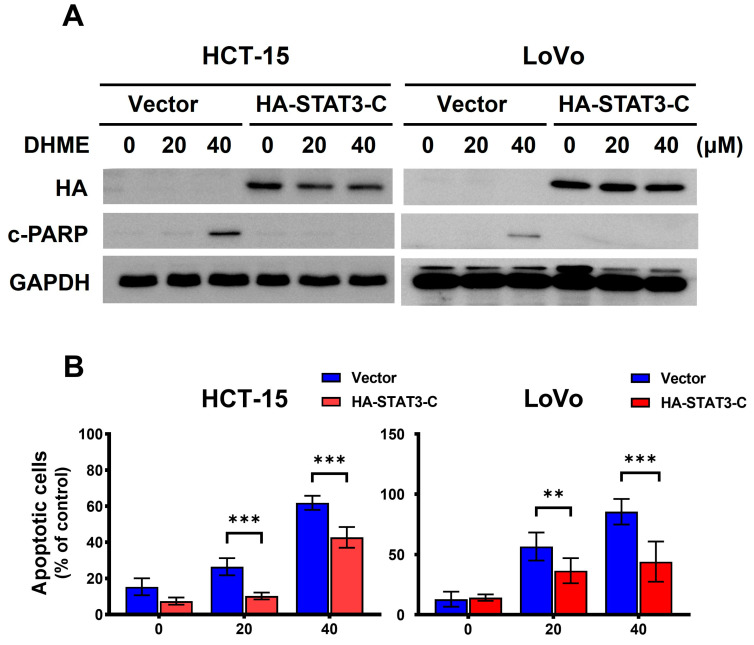
Essential role of STAT3 inhibition in the anti-CRC effect of DHME. (**A**,**B**) Sustained STAT3 activation blocks DHME-induced apoptosis. Human CRC cell lines stably expressing vector alone or a dominant-active *STAT3* mutant (STAT3 (A661C/N663C); STAT3-C) were treated with DHME (0, 20, 40 μM) for 24 h, followed by immunoblotting for the levels of cleaved PARP (c-PARP) (**A**) or by flow cytometry analysis for the levels of Annexin V-positive population (**B**) to define the extent of DHME-induced apoptosis. (**C**) Sustained STAT3 activation mitigates DHME-induced suppression of CRC colony-forming ability. CRC stable clones of vector control or STAT3-C were subject to clonogenicity assay to measure their cell viability following DHME treatment. GAPDH levels were used as equal loading control. ** *p* < 0.002; *** *p* < 0.001.

**Figure 4 biomedicines-11-02530-f004:**
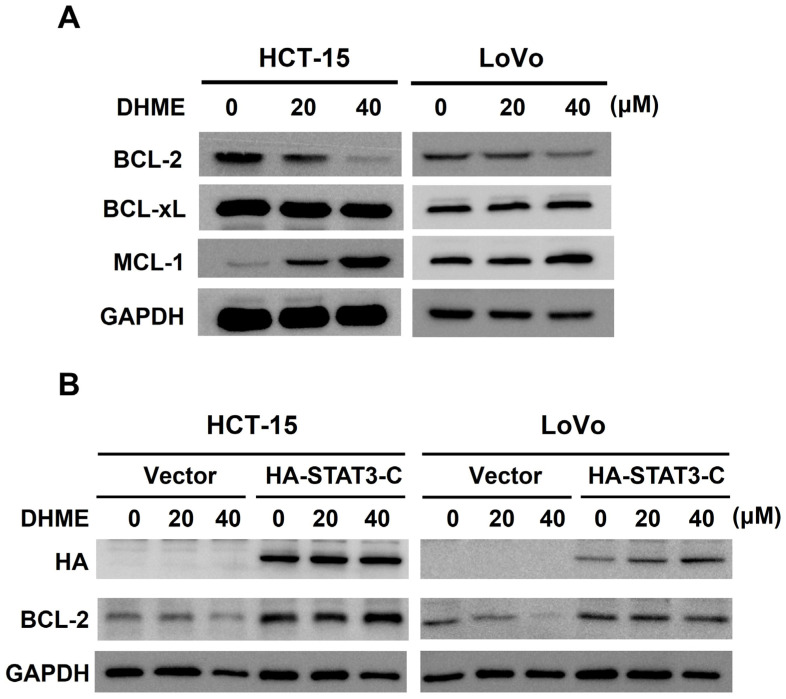
STAT3 inhibition accounts for DHME-induced downregulation of BCL-2 to elicit CRC cell apoptosis. (**A**) DHME downregulates BCL-2. Human CRC cells were treated with DHME (0, 20, 40 μM) for 24 h, followed by immunoblotting for the levels of BCL-2, BCL-xL, and MCL-1, all of them known as downstream effectors of STAT3. (**B**) DHME downregulates BCL-2 in a STAT3-dependent manner. DHME-treated CRC stable clones of STAT3-C or its vector control were subject to immunoblotting for the levels of BCL-2. (**C**,**D**) DHME downregulates BCL-2 for inducing CRC cell apoptosis. Human CRC stable clones of BCL-2 or its vector control were treated with DHME, followed by evaluating the extent of DHME-induced apoptosis using immunoblotting for the levels of cleaved PARP (c-PARP) (**C**) or Annexin V-positive cell population (**D**). ** *p* < 0.002; *** *p* < 0.001.

**Figure 5 biomedicines-11-02530-f005:**
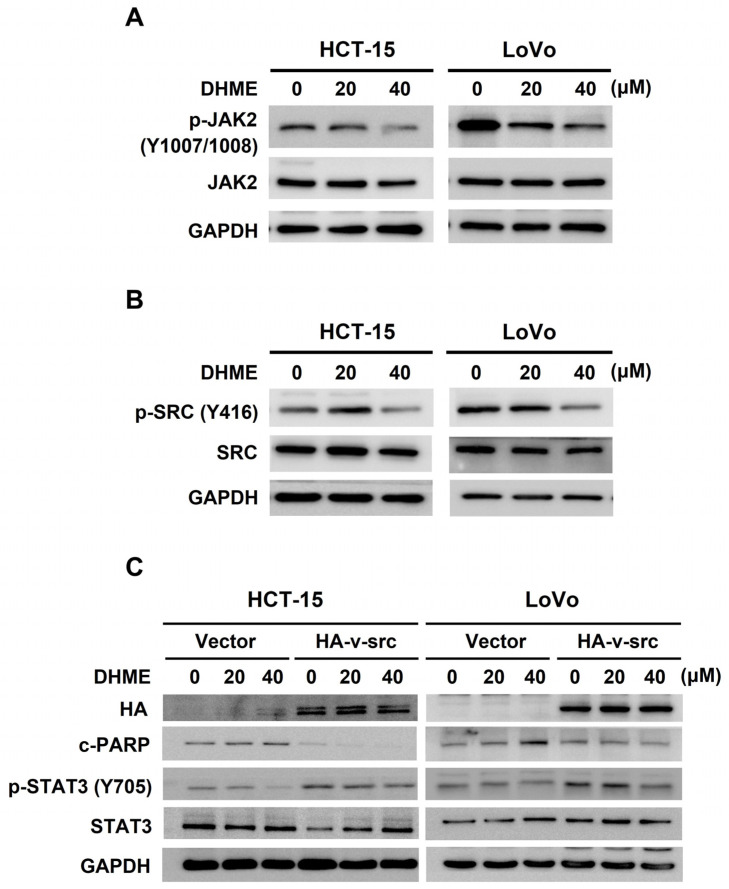
Inhibition of SRC activation underlies DHME-induced STAT3 blockade. (**A**) Limited effect of DHME on JAK2 activation. Human CRC cells treated with DHME (0, 20, 40 μM) were subjected to immunoblotting 24 h later for the amounts of tyrosine 1007/1008-dual phosphorylated JAK2 (p-JAK2 (Y1007/Y1008)), a surrogate marker of JAK2 activation. (**B**) DHME suppresses the activation of SRC. DHME-treated human CRC cells were subject to immunoblotting for the levels of tyrosine 406-phosphorylated SRC (p-SRC (Y406)), a substitute marker of SRC activation. (**C**) Sustained SRC activation thwarts DHME-induced blockade of STAT3 activation and apoptosis. Human CRC stable clones of v-src (a dominant-active SRC) or its vector control were subject to 24 h treatment with DHME and then were underwent immunoblotting for the levels of p-STAT3 (Y705) and cleaved PARP (c-PARP).

## Data Availability

Data will be available by corresponding author (chiachechang@gmail.com) upon reasonable request.

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
