# Peer review of "Blockade of the SRC/STAT3/BCL-2 Signaling Axis Sustains the Cytotoxicity in Human Colorectal Cancer Cell Lines Induced by Dehydroxyhispolon Methyl Ether"

_biomedicines, 2023, doi:10.3390/biomedicines11092530_

Round 1

Reviewer 1 Report

In this manuscript, the authors use dehydroxyhispolon methyl ether (DHME), obtained from the mushroom Phellinus linteus. This compound's potential is harnessed for the therapeutic intervention of colorectal cancer cell lines. The investigation is particularly oriented towards elucidating DHME's capacity to potentiate the efficacy of the prominent chemotherapeutic agent, 5-Fluorouracil, while concurrently displaying an absence of toxicity towards non-neoplastic cells. The current manuscript engenders great interest owing to its compelling narrative and holds promise as it showcases initial auspicious findings.

This manuscript presents a comprehensive and meticulously structured introduction, providing nuanced contextualization of the research area. The authors clarify the fundamental role of DHME and its pharmacological foundations within the field of colorectal cancer therapeutics. The methodological basis of the study is well established, being meticulous and precise, which translates into solid results.

However, it is critical to recognize that current research is predominantly focused on in vitro assessments of DHME bioactivity. While in vitro assays provide compelling information, the holistic validation of these findings could be significantly enhanced by an extension of the study to an in vivo setting. This fundamental progression would not only strengthen the reliability of the observed results but would also reinforce the credibility of the conclusions drawn from this study.

Reviewer 2 Report

The article entitled "Blockade of the SRC/STAT3/BCL-2 Signaling Axis Sustains the 2 Cytotoxicity in Human Colorectal Cancer Cell Lines Induced by Dehydroxyhispolon Methyl Ether, A Hispolon Analog” by Ya-Chu Hsieh et al., support the role of DHME as a effective and less toxic drug to induce apoptosis in CRC in vitro by blocking the SRC/STAT3/BCL-2 axis. The manuscript is well written and introduced and conclusions are supported by results.

Please find below my comments to improve the quality of the manuscript:

-I should change the title to include Dehydroxyhispolon Methyl Ether or a Hispolon Analog, but not both.

-Re-write mutant genes names and latin expressions in italics.

-All Western Blot from all figures must show values of the different bands by densitometry and express the value as mean ± standard deviation compared to GAPDH expression as housekeeping.

-Statistical analyses are not well performed since in those cases non-parametric test must be applied. Revise the analysis.

- DHME seems to enhance the expression of MCL-1 in a dose dependent manner; thus, efficacy of DHME downregulating p-STAT3 could be hampered by 5-FU resistance conferred by MCL-1 upregulation. Discussion of MCL-1 is rather limited specially the part refering to the effect of MCL1 overexpression, please discuss this finding.

It´s Ok 

Reviewer 3 Report

The findings appear to be interesting. Major points that the authors need to address are as follows:

1. The molecular mechanism by which DHME suppressed both constitutive and interleukin 6-induced STAT3 activation should be analysed in detail. For instance, can DHME directly bin dto STAT3 proteins and prevent their nuclear translocation or target activation of kinases involved in STAT3 activation.

2. Whether knockout of STAT3 using siRNA can abrogate the apoptotic effects of DHME in CRC cells? 

3. A limited in vivo study will greatly increase the impact of the findings.

4. The toxicity and pharmacokinetic profile of DHME should be analysed.

5. The manuscript should be carefully checked for typographical errors.

Moderate editing is required.

Round 2

Reviewer 2 Report

Thanks so much for the amended version of the manuscript.

Its OK.

Reviewer 3 Report

The authors have addressed all my concerns.

Not applicable.